# Epidemiology of thyroid disorders in the Lifelines Cohort Study (the Netherlands)

**Hanneke J. C. M. Wouters**[1], **Sandra N. Slagter**[1], **Anneke C. Muller Kobold**[2], **Melanie M. van der Klauw**[1], **Bruce H. R. Wolffenbuttel**[1] *

1 Department of Endocrinology, University of Groningen, University Medical Center Groningen, Groningen, The Netherlands, 2 Department of Laboratory Medicine, University of Groningen, University Medical Center Groningen, Groningen, The Netherlands

* bwo@umcg.nl

## Abstract

### Background

Thyroid hormone plays a pivotal role in human metabolism. In epidemiologic studies, adequate registration of thyroid disorders is warranted. We examined the prevalence of thyroid disorders, reported thyroid medication use, thyroid hormone levels, and validity of thyroid data obtained from questionnaires in the Lifelines Cohort Study.

### Methods

We evaluated baseline data of all 152180 subjects (aged 18–93 years) of the Lifelines Cohort Study. At baseline, participants were asked about previous thyroid surgery and current and previous thyroid hormone use. At follow-up (n = 136776, after median 43 months), incident thyroid disorders could be reported in an open, non-structured question. Data on baseline thyroid hormone measurements (TSH, FT4 and FT3) were available in a subset of 39935 participants.

### Results

Of the 152180 participants, mean (±SD) age was 44.6±13.1 years and 58.5% were female. Thyroid medication was used by 4790 participants (3.1%); the majority (98.2%) used levothyroxine, and 88% were females. 59.3% of levothyroxine users had normal TSH levels. The prevalence of abnormal TSH levels in those not using thyroid medication was 10.8%; 9.4% had a mildly elevated (4.01–10.0 mIU/L), 0.7% had suppressed (<0.40 mIU/L), while 0.7% had elevated (>10.0 mIU/L) TSH levels. Over 98% of subjects with TSH between 4 and 10 mIU/L had normal FT4. Open text questions allowing to report previous thyroid surgery and incident thyroid disorders proved not to be reliable and severely underestimated the true incidence and prevalence of thyroid disorders.

### Conclusions

Undetected thyroid disorders were prevalent in the general population, whereas the prevalence of thyroid medication use was 3.1%. Less than 60% of individuals using levothyroxine

**Data Availability Statement:** The manuscript is based on data from the Lifelines Cohort Study, Study OV15-0306. Lifelines adheres to standards for data availability. Due to ethical restrictions

imposed by the Lifelines Scientific Board and the Medical Ethical Committee of the University Medical Center Groningen related to protecting patient privacy, the data are not publicly available. The data catalogue of Lifelines is publicly accessible on https://www.lifelines.nl/researcher/data-and-biobank/$6102/$6104. All international researchers can obtain data at the Lifelines research office (research@lifelines.nl), for which a fee is required. The Lifelines system allows access for reproducibility of the study results.

**Funding:** The Lifelines Biobank initiative has been made possible by subsidy from the Dutch Ministry of Health, Welfare and Sport, the Dutch Ministry of Economic Affairs, the University Medical Center Groningen (UMCG the Netherlands), University Groningen and the Northern Provinces of the Netherlands. The current work was supported by the National Consortium for Healthy Ageing, and funds from the European Union's Seventh Framework program (FP7/2007-2013) through the BioSHaRE-EU (Biobank Standardisation and Harmonisation for Research Excellence in the European Union) project, grant agreement 261433. The funders had no role in study design, data collection and analysis, decision to publish, or preparation of the manuscript.

**Competing interests:** The authors have declared that no competing interests exist.

had a normal TSH level. The large group of individuals with subclinical hypothyroidism (9.4%) offers an excellent possibility to prospectively follow the natural course of this disorder. Both structured questions as well as linking to G.P.'s and pharmacists' data are necessary to improve the completeness and reliability of Lifelines' data on thyroid disorders.

## Introduction

Thyroid hormone plays a pivotal role in human metabolism. It is important in all processes in the body, including metabolic pathways, growth, development, cognition, energy homeostasis and temperature regulation. Peripheral thyroid hormone levels are closely regulated by the pituitary gland. Disorders of thyroid function are frequently diagnosed, with prevalences varying between 2 and 6% in large population-based studies [1–3].

Thyroid disorders and their treatment may have a great impact on long-term health. Thyrotoxicosis and even mild (subclinical) hyperthyroidism are associated with a higher risk of cardiovascular problems and osteoporosis [4–7], while hypothyroidism has been associated with dyslipidaemia, atherosclerosis and also an increased risk of cardiovascular events [8–11]. Over-substitution of levothyroxine in individuals with a thyroid disorder accounted for approximately half of both prevalent and incident suppressed thyroid stimulating hormone (TSH) findings in a community-based cohort, especially among older women, which increases their risk of atrial fibrillation and osteoporosis [12]. In order to assess the (life-time) consequences of mild thyroid dysfunction and thyroid disorders in epidemiologic studies, adequate registration of such disorders, the use of medication influencing thyroid function, and degree of control of thyroid dysfunction is necessary. Hypothyroidism may not be readily recognized, or remain even asymptomatic for a -yet unknown- period [3, 13], leading to a longer exposure to low thyroid hormone levels. More severe suppression of TSH in individuals treated for hypothyroidism and thyroid cancer has been associated with a higher incidence of cardiovascular events and osteoporosis [14, 15].

Biobanks offer the unique possibility to retrieve earlier serum samples to evaluate preceding thyroid hormone levels, especially when repeated evaluation and sample collection in their participants is performed. The Lifelines Cohort Study is such a biobank, examining in a unique three-generation design the health and health-related behaviours of the habitants in the North of the Netherlands. In this study, we examine the prevalence of thyroid disorders, thyroid hormone use, and thyroid hormone levels in the Lifelines Cohort Study. Results are compared with similar data obtained in other population-based studies and in the Dutch G.P. morbidity registry for validation purposes.

## Materials and methods

### Participants

Subjects included were participants in the Lifelines Cohort Study, a multi-disciplinary prospective population-based cohort study examining in a unique three-generation design the health and health-related behaviours of 167729 persons living in the North of The Netherlands. It employs a broad range of investigative procedures in assessing the biomedical, socio-demographic, behavioural, physical and psychological factors which contribute to the health and disease of the general population, with a special focus on multi-morbidity and complex genetics [16–18]. It was demonstrated that the Lifelines adult study population is broadly representative

for the adult population of the North of the Netherlands [19]. All participants have provided written informed consent before participating in the study, which has been approved by the Medical Ethics Review Committee of the University Medical Center Groningen.

For the present study, we evaluated the data of all 152180 subjects between 18 and 93 years of age who underwent a baseline examination between January 2007 and December 2013. In addition, follow-up data which have been collected between January 2014 and January 2017 were available for analysis in 136776 participants. Of these, 98626 had filled in follow-up questionnaires and underwent follow-up measurements, while in 38150 only interim questionnaire data were available. Median follow-up until the last questionnaire was 43 months (range 11 to 133 months).

Measurement of thyroid hormone levels, i.e. TSH, free thyroxine (FT4) and free triiodothyronine (FT3), started in November 2009. As Lifelines decided to stop the measurement of thyroid hormones in November 2011, these measurements were only available and complete in a subset of 39935 participants. There was no clinically relevant difference in sex distribution, age and body mass index (BMI) between those with and without thyroid hormone levels measured. In 37741 of these 39935 participants, follow-up data were available. Of these, 30050 had filled in follow-up questionnaires and underwent follow-up measurements, while in 7691 only interim questionnaire data were available. Median follow-up in this subgroup was 48 months (range 12 to 112 months).

## Questionnaires and clinical examination

The specific methodology employed by Lifelines has been described previously [17, 20, 21]. At both baseline and follow-up examination, participants completed a self-administered questionnaire on medical history and past and current diseases. The catalogue of these questionnaires can be browsed at https://www.lifelines.nl. A limited number of participants (3%) was not able to fill in the questionnaires themselves; for those participants a proxy questionnaire was used and filled in by a close relative. The proxy questionnaire was considerably shorter than the full questionnaires. Questions related to thyroid disorders have been summarised in Table 1. During the baseline examination, participants were asked about current and previous use of medication for a thyroid disorder (HEALTH66/67), and concomitant thyroid disorders (HEALTH73). Previous thyroid surgery (HEALTH74) could be reported by participants in an open, non-structured question. During follow-up, no structured questions regarding thyroid disorders or medication use were asked, therefore *incident* thyroid disorders were based on self-report obtained from an open, non-structured question (HEALTH100).

**Table 1. Content of the questionnaires.**

| Question# | Description in the questionnaire | |
|---|---|---|
| Baseline | | |
| HEALTH66 | Do you currently use medication for an overactive or underactive thyroid? | Yes / No |
| HEALTH67 | Have you used medication for an overactive or underactive thyroid in the past? | Yes / No |
| HEALTH73 | Do you have or have you had another disorder that you have not mentioned yet? | Yes / No |
| | If you have had another disorder, what disorder? | Open text answer |
| HEALTH74 | If you ever had surgery, what was the reason for the surgery? | Open text answer |
| Follow-up# | | |
| HEALTH100 | Could you indicate which other health problems you have (had) since the last time you filled out this questionnaire? | Open text answer |

\# once every 1.5 years.

Medication use was verified at baseline by a certified research assistant, and scored by the Anatomical Therapeutic Chemical (ATC) Classification System, a system developed for the classification of active ingredients of drugs. We discriminated between the use of levothyroxine (ATC code H03AA01), liothyronine (ATC H03AA02), and thyroid blockers (methimazol or propylthiouracil, ATC code H03BA and H03BB). For most evaluations, we excluded 60 participants using amiodarone (ATC code C01BD01), as this drug may significantly alter thyroid hormone levels [22]. Structured ascertainment of thyroid disorders or medication use by Lifelines participants with the use of other sources, like general practitioners' or pharmacists' data, was not performed.

A standardised protocol was used to obtain blood pressure and anthropometric measurements: height, weight, and waist circumference. Weight was measured to the nearest 0.1 kg and height and waist circumference to the nearest 0.5 cm by the research assistant using calibrated measuring equipment, with participants wearing light clothing and no shoes. Waist circumference was measured with a tape around the body between the lower rib margin and the iliac crest. BMI was calculated as weight divided by height squared (kg/m$^2$). Systolic and diastolic blood pressure (BP) and heart rate were measured every minute for 10 minutes in the supine position using an automated Dinamap Monitor (GE Healthcare, Freiburg, Germany). The average of the last three readings was recorded for each blood pressure parameter and heart rate.

## Prevalence and incidence comparison with other sources

For comparison with Lifelines epidemiologic data, we have assessed the prevalence of hypothyroidism (International Classification of Primary Care (ICPC) T86) as collected by NIVEL (Netherlands Institute for Health Services Research) through registration by G.P.'s. This public dataset was created in March 2014, and comprised data on incidence and prevalence of hypothyroidism during the year 2013.

Our second source for comparison was NHANES, a cross-sectional survey in the USA that uses a complex, stratified, multistage sampling design [23, 24]. We used data from NHANES 2007–2008, 2009–2010 and 2011–2012 participants who were aged 18 years and above, and had serum TSH measurements available (n = 9209). Written informed consent was obtained from all participants; the survey protocol was approved by the Research Ethics Review Board of the National Center for Health Statistics. Interviews, examination response rates, methodology and results are publicly available [25]. Hypothyroidism was defined as the use of levothyroxine.

## Biochemical measurements

Blood samples were taken in the fasting state between 8:00 and 10:00 a.m. and transported to the laboratory facility at room temperature or 4˚C, depending on the sample requirements. All measurements were performed the same day. Levels of TSH, FT4 and FT3 were assayed by electrochemiluminescence immunoassay (Roche Modular E170, Roche, Switzerland). Glycated haemoglobin (HbA1c, EDTA-anticoagulated) was analysed using an NGSP-certified turbidimetric inhibition immunoassay on a Cobas Integra 800 CTS analyser (Roche Diagnostics BV, Almere, The Netherlands). Serum creatinine was measured on a Roche Modular P chemistry analyser (Roche, Basel, Switzerland). Total and high density lipoprotein (HDL) cholesterol were measured using an enzymatic colorimetric method, triglycerides using a colorimetric UV method, and low density lipoprotein (LDL) cholesterol using an enzymatic method, on a Roche Modular P chemistry analyser (Roche, Basel, Switzerland). Fasting blood glucose was measured using a hexokinase method. The general Dutch population is iodine sufficient [26]. Anti-thyroid peroxidase antibody levels were not available.

## Definitions of concomitant morbidities, calculations, and statistical analyses

Diagnosis of metabolic syndrome was established if a subject satisfied at least three out of five criteria according to the modified guidelines of the National Cholesterol Education Programs Adults Treatment Panel III (NCEP ATPIII criteria): 1. systolic blood pressure $\geq$130 mmHg and/or diastolic blood pressure $\geq$85 mmHg and/or use of antihypertensive medication; 2. HDL cholesterol levels <1.03 mmol/L in men and <1.30 mmol/L in women and/or use of lipid-lowering medication influencing HDL levels; 3. triglyceride levels $\geq$1.70 mmol/L and/or use of triglyceride-lowering medication; 4. waist circumference $\geq$102 cm in men and $\geq$88 cm in women; 5. fasting glucose level $\geq$ 5.6 mmol/L and/or use of blood glucose-lowering medication and/or diagnosis of type 2 diabetes [27]. Diagnosis of type 2 diabetes was by self-report, use of oral blood-glucose-lowering agents or insulin, or a fasting blood glucose $\geq$ 7.0 mmol/L, and/or a HbA1c $\geq$ 6.5% at baseline laboratory evaluation. Use of statins and blood-pressure-lowering drugs was assessed based on the ATC codes of verified medication use (statins: ATC code C10AA**; BP-lowering drugs: ATC codes C02*, C03*, C04*, C07*, C08*, C09*).

We cross-tabulated verified medication use with the answers to questions HEALTH66 and HEALTH67 (self-reported current and previous use of medication for a thyroid disorder, Table 2). The percentage of users of thyroid hormone medication were calculated per age group for the entire cohort. In those participants with available thyroid hormone measurements, mean levels of TSH, FT4 and FT3 were calculated according to the use of levothyroxine. For these calculations, participants using liothyronine and thyroid hormone blockers were excluded. Participants were divided into subgroups according to serum TSH values (1st: suppressed, TSH <0.40 mIU/L; 2nd: euthyroid, TSH 0.40–4.0 mIU/L (reference group); 3rd: mildly elevated, TSH 4.01–10.0 mIU/L, also defined as subclinical hypothyroidism; 4th: markedly elevated, TSH >10.0 mIU/L, overt hypothyroidism). The percentage of users of thyroid hormone and distribution of TSH class was calculated for all age-decade groups (18–29, 30–39, 40–49, 50–59, 60–69, 70–79, $\geq$80 years). For comparison of Lifelines with NIVEL and NHANES data, the prevalence of thyroid hormone use was calculated for the age groups 18–44, 45–64, 65–74, 75–84, and $\geq$85 years, since these were the age groups used by NIVEL. Local laboratory reference values for FT4 were 11–20 pmnol/L, for FT3 4.4–6.7 pmol/L.

All answers to the open questions HEALTH73, HEALTH74 and HEALTH100 which were related to thyroid disorders were scored according to the most likely diagnosis or reason for surgery: 1. Hypothyroidism; 2. Hyperthyroidism not specified; 3. Graves' disease; 4. Nodular disease or goiter; 5. Thyroid cancer; 6. Thyroiditis; 7. Thyroid disorder not specified.

**Table 2. Current use of medication for a thyroid disorder (question HEALTH66).**

|  | Yes | No | Missing | Total |
|---|---|---|---|---|
| Are you currently using medication for a thyroid disorder? | 4579 (3.0%) | 143870 (94.5%) | 3731 (2.5%) | 152180 |
| Cross-check with medication overview list |  |  |  |  |
| Levothyroxine | 4416 (96.4%) | 56 (<0.1%) | 231 (6.3%) | 4703 |
| Liothyronine | 68 (1.5%) | <10 (<0.1%) | 0 (0%) | n.a. |
| Thyroid blocker | 106 (2.3%) | <10 (<0.1%) | <10 (<0.1%) | n.a. |
| Animal source of thyroid hormone | 19 (0.4%) | <10 (<0.1%) | <10 (<0.1%) | n.a. |
| Any of the above* | 4491 (98.1%) | 63 (0.04%) | 236 (6.3%) | 4790 |
| No medication used / reported | 88 (1.9%) | 143807 (>99%) | 3495 (93.7%) | 147390 |

* a participant could use one or more of these medications.

All analyses were conducted using SPSS Statistics (Version 22, IBM, Armonk, NY, USA). Data are presented as mean ± SD, or median and IQR (interquartile range) when not normally distributed, or percentages. Means were compared between groups with analysis of variance. When variables were not normally distributed, medians were compared with the nonparametric Mann-Whitney U test. Chi-square test was used to analyse categorical variables. P-values < 0.05 were considered statistically significant.

## Results

### Baseline dataset

S1 Table shows the most relevant baseline parameters for the participants according to levothyroxine use. Of the 152180 participants, 89050 (58.5%) were female. Mean age was 44.6 (SD 13.1, range 18–93) years, and mean BMI was 26.1 (SD 4.3, range 13.4–73.6) kg/m². Participants reporting the use of thyroid hormone were older, had a higher BMI, and were more frequently female and obese (BMI >30 kg/m²). They were also less healthy as they had higher glucose, HbA1c and total cholesterol and triglyceride levels, more frequently had metabolic syndrome or type 2 diabetes, and used a higher number of medications including statins and antihypertensive medication.

### Thyroid disease based on current and previous medication use

The question on current use of medication for an underactive or overactive thyroid was answered positively by 4579 participants (3.0% of total). The majority of them (96.4%) reported the use of levothyroxine (Table 2). Of those reporting no use of medication, 63 (0.04%) did use thyroid medication according to the verification of the medication list, again the majority using levothyroxine. An answer to this question was missing in 3731 participants, as this question was not part of the proxy questionnaire. Still, 236 (6.3%) of them were using medication according to the verified medication list, bringing the total users of medication to 4790 (= 3.1% of the total population). In total 118 participants (0.08%) used a thyroid blocker indicating active treatment for thyrotoxicosis.

A total of 3169 participants reported at baseline that they have used medication for an underactive or overactive thyroid in the past. Of these, 1800 (56.8%) were still using any type of thyroid medication at the baseline visit according to the verified medication list. This indicates the low discriminatory power of this question to indicate earlier but not current use of any thyroid medication. Fig 1 shows the distribution of levothyroxine use per age group. There is a clear increase with age, and for all age groups a 1:5 to 1:6 male:female ratio. The overall prevalence of levothyroxine use was 3.1% (0.9% in men and 4.7% in women).

In Table 3, we compared the incidence and prevalence of hypothyroidism (ICPC code T86) of the NIVEL G.P. registration with the prevalence of hypothyroidism (defined as levothyroxine use) in Lifelines. The prevalence of hypothyroidism in male participants was similar, but in females the prevalence of hypothyroidism appeared to be higher in Lifelines for almost all age groups, except those >85 years. In addition, we compared our data to NHANES. In NHANES the prevalence of hypothyroidism (defined as any use of thyroid hormone) was higher than that in Lifelines, especially for individuals >45 years in both males and females.

### Thyroid hormone levels

In 39935 participants, thyroid hormone levels were available. For the subsequent analyses, we excluded users of methimazole, propylthiouracil, animal source of thyroid hormone and/or

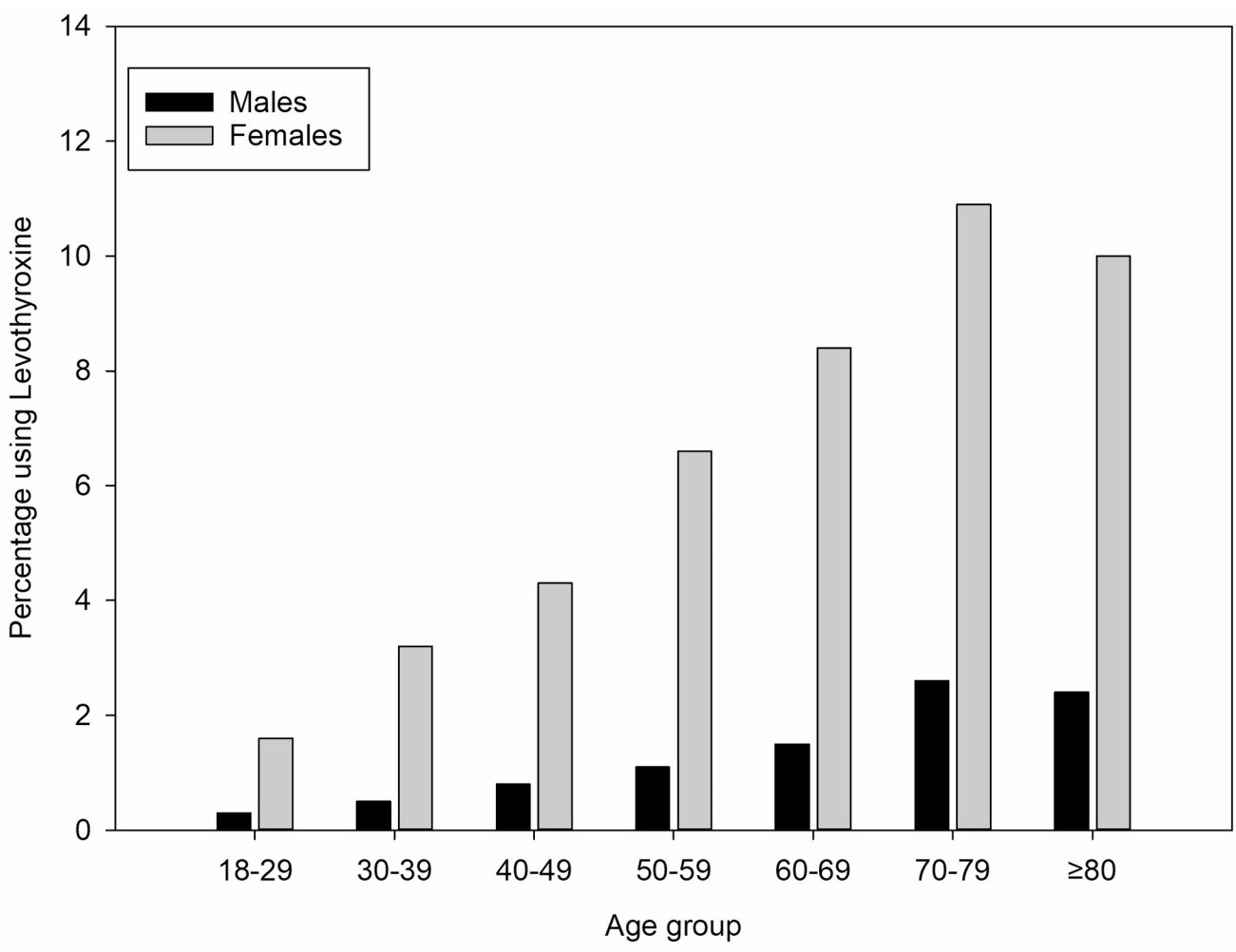

**Fig 1. Percentage of levothyroxine users per age group.**

**Table 3. Incidence and prevalence of hypothyroidism (in number per 1000).**

| | | Incidence NIVEL[a] | | Prevalence NIVEL [a] | | Prevalence NHANES [b] | | Prevalence Lifelines [c] | |
|---|---|---|---|---|---|---|---|---|---|
| | **Total** | **4.3** | | **21.8** | | **unknown** | | **unknown** | |
| **Age group** | | **Males** | **Females** | **Males** | **Females** | **Males** | **Females** | **Males** | **Females** |
| 0–4 yrs | | 0.3 | 0.1 | 1 | 0.6 | NA | NA | NA | NA |
| 5–17 yrs | | 0.3 | 0.8 | 1 | 2.6 | NA | NA | NA | NA |
| 18–44 yrs | | 0.9 | 5.1 | 3.5 | 22.1 | 3.5 | 31.0 | 5.1 | 29.8 |
| 45–64 yrs | | 2.3 | 9.3 | 10.3 | 51.4 | 25.0 | 113.1 | 10.5 | 58.7 |
| 65–74 yrs | | 4.2 | 123 | 18.4 | 74 | 49.0 | 118.7 | 17.8 | 95.3 |
| 75–84 yrs | | 5.4 | 14.9 | 26.4 | 77.5 | 112.5 | 214.0 | 30.9 | 113.7 |
| ≥85 yrs | | 8.5 | 14.8 | 30.5 | 73.4 | NA | NA | 28.2 | 60.8 |

Sources:

[a] Open access NIVEL Registration of Care by G.P.'s, accessed March 2014, based on diagnosis code T86, registered by a G.P.

[b,c] Based on use of thyroid hormone supplementation.

liothyronine (n = 76). Participants using levothyroxine had higher FT4 and lower FT3 compared to those not using levothyroxine, with comparable TSH concentrations (S2 Table).

As shown in Table 4, prevalence of abnormal thyroid hormone levels in those not using thyroid medication was 10.8%; 9.4% had a mildly elevated TSH level (4.01–10.0 mIU/L, subclinical hypothyroidism), 0.7% had a suppressed TSH level (<0.4 mIU/L), while 0.7% had elevated TSH (>10 mIU/L, overt hypothyroidism). Fig 2 shows the inverse relationship between TSH and FT4 levels in individuals with elevated TSH (>4.0 mIU/L) not using levothyroxine. In total, 3600 of 3660 participants (98.4%) with TSH between 4.01 and 10.0 mIU/L still had a normal FT4 (>11.0 pmol/L). Of those with TSH >10.0 mIU/L, 61% had FT4 levels within the reference range. In participants using levothyroxine, less than 60% had TSH levels within the reference range of 0.40–4.0 mIU/L, 13.7% had suppressed TSH, while 27.0% had elevated TSH levels (Table 4). Prevalence of suppressed TSH gradually increased with increasing age in non-users (Table 5). With the exception of the high prevalence of subclinical hypothyroidism in those aged 18–29 years (13%), there is also a gradual increase of prevalence of subclinical hypothyroidism with ageing. For comparison, S3 Table shows similar data for NHANES. Compared to Lifelines, NHANES observed an overall higher prevalence of suppressed TSH, whereas the percentage of individuals with mildly or severely elevated TSH was lower.

## Validation of self-reported thyroid disorders and / or surgery

A total of 282 participants answered in question HEALTH73 that they had a thyroid disorder (S4 Table). Only 49 of them had answered 'yes' to either question HEALTH66 or HEALTH67 related to medication use for an overactive or underactive thyroid. The majority reported hypothyroidism. Considering the large difference between this question HEALTH73 and questions HEALTH66/67, these data must be considered as incomplete. A total number of 674 participants (0.4%) reported in the open question on surgery (HEALTH74) that they previously had undergone thyroid surgery (S4 Table). Over 50% reported no specific reason for their surgery.

S5 Table shows the number of participants reporting a new thyroid disorder in one of the follow-up questionnaires (question HEALTH100). Hypothyroidism and an unspecified thyroid disorder (the participant did not specify the nature of the thyroid disorder) were reported most frequently. Based on the number of participants and the duration of follow-up, this would suggest an incidence of hypothyroidism between 1.6 and 2.3 per 1000 individuals per year, and a total incidence of thyroid disorders of 3.1 per 1000 individuals per year. These incidence data are considerably lower than the incidence of hypothyroidism as collected by Dutch G.P.'s (Table 3). Unfortunately, the development of new thyroid disorders reported by Lifelines participants could not be checked with G.P. datasets or new medication use. However, in the subgroup of participants with available TSH levels at baseline, we were able to ascertain

**Table 4. Distribution of TSH class in participants according to levothyroxine use*.**

|  | No levothyroxine | | | Levothyroxine | | |
|---|---|---|---|---|---|---|
|  | **All** | **M** | **F** | **All** | **M** | **F** |
| TSH <0.4 mIU/L | 265 (0.7%) | 91 | 174 | 155 (13.7%) | 17 | 138 |
| TSH 0.4–4.0 mIU/L | 34545 (89.2%) | 14825 | 19720 | 670 (59.3%) | 68 | 602 |
| TSH 4.01–10.0 mIU/L | 3660 (9.4%) | 1169 | 2491 | 254 (22.5%) | 31 | 223 |
| TSH ≥10.0 mIU/L | 266 (0.7%) | 77 | 189 | 51 (4.5%) | 12 | 39 |
| Total (n) | 38736 | 16162 | 22574 | 1130 | 128 | 1002 |

*Participants reporting the use of thyroid blockers (methimazole or propylthiouracil), liothyronine, animal source thyroid hormone and amiodarone were excluded.

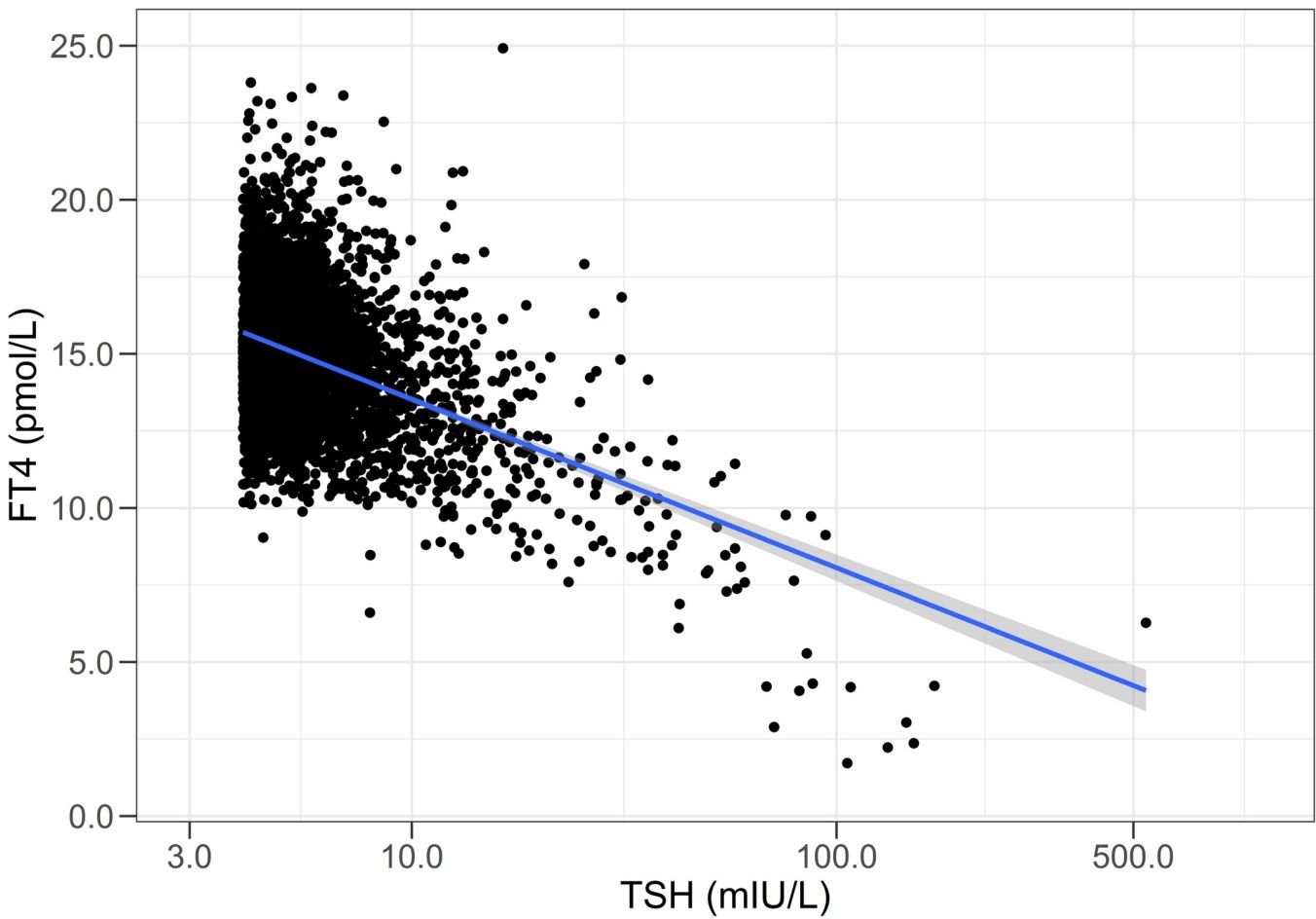

**Fig 2. Relationship between TSH and FT4 in participants with subclinical or overt hypothyroidism not using levothyroxine.** The line indicates the linear regression relationship. Note that 98.4% of the individuals with a TSH between 4.01 and 10.0 mIU/L had a normal FT4 concentration (>11 pmol/L).

how many individuals with a substantially elevated TSH level at baseline reported a new thyroid disorder during one of the follow-up questionnaires. Of the 70 participants with TSH level > 20.0 mIU/L without using thyroid hormone at baseline, only 18 reported the development of a new thyroid disorder. Reversely, we calculated that of 236 participants who reported a new hypothyroidism during follow-up, 119 (50.4%) were already using thyroid hormone supplementation at baseline.

**Table 5. Distribution of TSH class per age group (in %) in participants not using thyroid medication or amiodarone at baseline.**

| Age group (years) | N | TSH levels (mIU/L) | | | |
|---|---|---|---|---|---|
| | | <0.40 | 0.40–4.0 | 4.01–10.0 | >10.0 |
| 18–29 | 5111 | 0.4 | 86.1 | 13.0 | 0.5 |
| 30–39 | 8017 | 0.5 | 92.2 | 6.7 | 0.6 |
| 40–49 | 13365 | 0.7 | 91.0 | 7.7 | 0.6 |
| 50–59 | 6793 | 0.7 | 88.0 | 10.3 | 1.0 |
| 60–69 | 4253 | 1.1 | 84.9 | 13.2 | 0.8 |
| 70–79 | 1084 | 1.1 | 84.0 | 13.9 | 1.0 |
| ≥80 | 113 | 6.2 | 77.0 | 16.8 | 0.0 |

## Discussion

### Prevalence of thyroid disorders based on use of thyroid medication

In the Lifelines Cohort Study, we showed that the overall prevalence of diagnosed and treated thyroid disorders (hypothyroidism plus thyrotoxicosis), assessed by medication use, is 3.1%. Compared to the NIVEL G.P. registration, the prevalence of hypothyroidism (defined as levothyroxine supplementation) in male Lifelines participants was similar. In females the prevalence of hypothyroidism appeared to be higher. Thyroid hormone measurements in a subset of almost 40000 participants showed that the prevalence of unknown / undetected subclinical hypothyroidism was 9.4%, while the prevalence of undetected overt hypothyroidism was 0.7%. As expected, the use of levothyroxine increased with age. However, only 59.3% of users of levothyroxine had a TSH level within the normal range, meaning that possibly a large proportion is being undertreated (TSH >4.0 mIU/L, 27%) or overtreated (TSH<0.4 mIU/L, 13.7%).

In comparison with the NIVEL G.P. registrations, the estimated prevalence of hypothyroidism in Lifelines was similar in men, but in women it was higher except in those >85 years. This difference may be a consequence of the different way how hypothyroidism was defined, i.e. ICPC T86 coding (NIVEL) versus use of levothyroxine (Lifelines). Also, the time period in which data have been collected differed slightly, i.e. 2007–2013 in Lifelines, and 2013 in the NIVEL / G.P. registration. It should be noted that there is very limited information on how well hypothyroidism is diagnosed and scored in general practice. Research by NIVEL indicated that slightly over 80% of patients using thyroid hormone had a concomitant T86 ICPC code (registered hypothyroidism / myxoedema) [28]. The assumption that over 90% of levothyroxine users have this medication prescribed because of primary hypothyroidism may lead to the conclusion that there is a 10% underreporting. In addition, a recent report on the consequences of the temporary withdrawal of the Thyrax[R] brand from the Dutch market mentioned that about 8% of levothyroxine users only used a dose of 25 mcg per day or less [29], suggesting that a considerable number of patients labelled with ICPC code T86 had subclinical and not overt hypothyroidism. This may also apply to the Lifelines dataset.

### Prevalence and incidence of thyroid disorders based on thyroid hormone levels

In Lifelines participants, a new diagnosis of overt hypothyroidism (TSH >10.0 mIU/L) was made in 0.7%, while the prevalence of subclinical hypothyroidism was 9.4%. A significant number (61%) of participants with TSH >10.0 mIU/L still had normal FT4 levels. Over the previous decades, the (reported) prevalence of thyroid disease has changed. One of the first studies to evaluate thyroid function on a large scale in the population was the Whickham Survey, which was conducted between 1972 and 1974 [30]. The prevalence of established hypothyroidism was 1.4% in women and 0.1% in men. Five new cases of overt hypothyroidism were found (0.35%), while prevalence of elevated TSH (defined as ≥6.0 mIU/L) was 7.5% in women and 2.8% in men [30]. It must be realized that in those days, treatment was different (thyranon, thyroid powder), and immunoassays were less developed. In 1985, the Framingham study reported that the prevalence of hypothyroidism was 4.4% (95/2139 participants over the age of 60 years), with a higher prevalence in women (5.9%) compared to men (2.4%). Prevalence of subclinical hypothyroidism (defined as TSH 5.0–10.0 mIU/L) was 5.9% [31]. In a follow-up study of the Whickham Survey, the incidence of hypothyroidism was calculated to be 3.5 per 1000 per year in women and 0.6 per 1000 per year in men [32]. Other studies, summarized by Vanderpump in an extensive review in 2011, reported a prevalence of previously undiagnosed overt hypothyroidism between 0 and 7.8 per 1000 men, and 3.0 and 20.5 per 1000 women [33].

Data from 1995–1997 in Norway were similar, with a prevalence of treated hypothyroidism of 4.8% in women and 0.89% in men, and a clear increase of prevalence with higher age [34]. In addition, the prevalence of undiagnosed hypothyroidism (TSH $\geq$10.0 mIU/L) was 0.9% in women and 0.37% in men, while the prevalence of subclinical hypothyroidism was 5.1 and 3.7%, respectively [34]. In a follow-up study between 2006–2008 from HUNT, the prevalence of undiagnosed overt as well as subclinical hypothyroidism had declined considerably, while the prevalence of treated hypothyroidism had increased from 5.0 to 8.0% in women, and from 1.0 to 2.0% in men [2]. The explanation of the investigators for these changes were increased thyroid function testing as well as earlier levothyroxine treatment for subclinical hypothyroidism [2]. A similar trend was observed between 2000 and 2010 surveys in Pomerania [35]. This may suggest that a significant number of individuals is treated with thyroid hormone supplementation without possible medical necessity [36]. Over 60% of individuals in whom subclinical hypothyroidism is established, thyrotropin declines to the normal range over 5 years [36]. Too liberal thyroid hormone treatment in individuals with moderately elevated TSH levels may be one of the factors responsible for differences in medication use between studies. This trend could also be observed in the Thyroid Epidemiology Audit and Research Study (TEARS) in Tayside, UK, where almost half of the individuals who started taking thyroid hormone supplementation had a TSH level <6.0 mIU/L [37]. Similar mechanisms may explain the higher prevalence of levothyroxine use between NHANES and Lifelines (Table 3).

Other reasons for the difference in prevalence may be differences in reference values. One study in Korea based its reference values for TSH, measured with similar methodology as in our present study, on the 2.5th and 97.5th percentile of TSH values in subjects with no prior history of thyroid disease, no family history of disease, negative TPO-antibodies, and serum FT4 levels in the reference range [38]. This yielded a reference value for TSH from 0.62 to 6.86 mIU/L, which significantly lowers the percentage of individuals with subclinical hypothyroidism compared to our reference values. A study in France suggested the use of age-specific and sex-specific reference values for TSH, with upper-limit reference values >5.0 mIU/L for women over the age of 70 [39]. This might partly explain the large number of individuals with a normal FT4 in combination with a slightly elevated TSH in the current study.

## Prevalence of abnormal TSH levels in individuals using levothyroxine

Previous research has clearly shown that -with TSH levels in the normal range- FT4 levels were significantly higher and yet FT3 levels were significantly lower (p<0.001 in both cases) in levothyroxine-treated athyreotic patients than in matched euthyroid control subjects [40]. In Lifelines participants using levothyroxine, only 59.3% had TSH levels within the normal range, while 27% had a TSH > 4.0 mIU/L and appeared to be undertreated, and 13.7% had a suppressed TSH level and was possibly overtreated. Similarly, in 2000 the Colorado Thyroid Disease Prevalence Study reported that 60% of individuals using thyroid medication had a normal TSH level [41]. The same study reported that the percentage of participants taking thyroid medication with subclinical hyperthyroidism (TSH between 0.01–0.3 mIU/L) was 20%, while 17% had subclinical hypothyroidism (TSH >5.1 mIU/L and total T4 >57.9 nmol/L) [41]. Similar data were reported in the UK and Brasil, with the latter study showing poorer health-related quality of life associated with undertreatment [42, 43]. A recent study in The Netherlands evaluating a levothyroxine brand switch amongst individuals with hypothyroidism treated in general practice showed that 67% of participants had a TSH level within the reference range [29]. The reasons for this are not clear. On the one hand, doctors may ignore the significance of elevated TSH levels when concomitant measurements show an FT4 level within the normal range. On the other hand, it may well be that despite an elevated TSH,

levothyroxine dose is not increased when patients do not report any complaints. TSH levels may vary over time, however, and a G.P. may decide not to alter the dose when the previous TSH level was within the target. The Dutch guidelines for G.P.'s nevertheless mention that with optimal supplementation, TSH is in the low-normal range of 0.5–2.0 mIU/L, and FT4 then is 'high-normal' [44].

## Validity of the obtained thyroid data

The extensive set of data collected by Lifelines supports its merits for thyroid-oriented research. Previously we have reported in smaller-sized studies based on baseline Lifelines data on the relationship between thyroid hormone levels and health-related quality of life [13], and metabolic syndrome [18], and the effect of the Thr92Ala polymorphism on thyroid function and health-related quality of life [21]. Also, in Lifelines genome-wide association study (GWAS) studies have been performed in over 13000 participants, and Lifelines data have contributed to several genetic evaluations including those related to thyroid function [45, 46].

Because of the longitudinal design, the Lifelines study may offer additional opportunities to expand our knowledge on the natural course of (subclinical) hypothyroidism and its complications. We observed a 0.7% prevalence of elevated TSH >10 mIU/L within the subset of Lifelines' participants not using thyroid medication with thyroid function measured. This implies that approximately 800 of the 112000 Lifelines participants in whom thyroid hormone was not measured, a TSH level >10.0 mIU/L could have been found. At follow-up, at least 778 new diagnoses of hypothyroidism and 317 unspecified thyroid disorders have been reported by participants (S5 Table). Also, 3660 participants had a TSH level between 4.0 and 10.0 mIU/L. Considering the current collection of information of Lifelines participants, these further investigations should be done in substudies in which additional data are collected, and verification of diagnosis and medication through G.P.'s charts and pharmacists' prescription data is indicated. Also, to better classify the relevance of subclinical hypothyroidism, measurement of anti-TPO antibodies would be of value. Currently such connections are being made, bringing an integrated big data approach where biobanks and clinical datasets are connected [47].

At baseline, one of the most important sources for establishing diagnoses and co-morbidities was the list of medication used by a participant. Medication has been shown to be a powerful indicator for (co-)morbidity [48, 49], health-related quality of life, and even severity of disease [50, 51]. Also the studies in Tayside, UK, considered medication use as the most important source for classifying participants with thyroid dysfunction [37]. Unfortunately, no data are available on the use of new medications or changes in medication at follow-up in Lifelines. Medication use like levothyroxine or methimazole can validate the self-reported diagnosis of hypothyroidism and thyrotoxicosis, or even ascertain the presence of a thyroid disorder when a participant did not report this correctly in the questionnaire. However, it must be noted that thyroid hormone supplementation is prescribed both in case of primary and secondary hypothyroidism. From the current dataset, we could not differentiate between these two causes, although epidemiologic data suggest that the majority of levothyroxine use is for primary hypothyroidism, either Hashimoto's hypothyroidism or hypothyroidism after treatment of thyrotoxicosis with surgery or radioactive iodine. Concomitant use of for instance hydrocortisone, sex hormones like testosterone and / or growth hormone could suggest the presence of secondary hypothyroidism. However, pituitary insufficiency comes in many variants, and also primary hypothyroidism and primary adrenal insufficiency (in case of hydrocortisone use) do co-exist. Adequate differentiation between both variants of hypothyroidism should therefore also be done by verification of G.P.'s charts.

## Strengths and limitations

Our study has several strengths and weaknesses. We presented baseline data on 152180 participants within a broad range of age and comorbidities, and the results of thyroid function in one of the largest population-based thyroid hormone screenings available, with extensive phenotyping in several domains of health and disease. Limitations are mainly found in the questionnaires. The Lifelines open text questions at baseline allowed participants to report whether they had undergone thyroid surgery, and the follow-up questionnaires allowed to report whether they had developed a new thyroid disorder. Both proved not to be reliable and severely underestimate the true incidence and prevalence of thyroid disorders. The number of answers to the question HEALTH73 (Do you have or have you had another disorder that you have not mentioned yet?) is only a fraction of the number of individuals who were using thyroid hormone supplementation. By extrapolation, we assume that the same holds true for the question regarding previous surgery for a thyroid disorder (HEALTH74). The majority of answers (55%) just mentioned thyroid surgery without any details on cause or type of procedure. In agreement with advices described in a 2009 editorial [52], the Lifelines questionnaires specifically ask for the history of several diseases as recognized and recalled by the patient, for instance several types of cardiovascular disease, osteoporosis, diabetes, hypertension, cancer, rheumatoid arthritis and inflammatory bowel diseases. However, the only specific questions asking for a thyroid disorder are the questions on current (HEALTH66) and past (HEALTH67) use of medication for a thyroid disorder. Even when such a specific question is introduced in future questionnaires, we can discuss how well they will be answered. Over half of the participants who reported earlier use of thyroid medication still was using medication at baseline screening. In addition, 10% of participants who answered that they underwent thyroid surgery, reported a cyst as a reason for surgery, which is not current practice in our country. Also, only a limited number of the participants who were found to have severely elevated TSH levels at baseline screening, reported a new thyroid disorder in one of the follow-up questionnaires. It should be noted that all TSH levels >10.0 mIU/L were communicated with the G.P. of a participant, with the advice (depending on the level of TSH) to re-check if needed, and otherwise to start levothyroxine supplementation. As mentioned earlier, validation of diagnoses and previous surgical procedures can be realised by exchanging data with charts of G.P.'s practices, and by collecting surgical notes from hospital charts. Lifelines is in an excellent position to increase its value and significance this way, especially because all health-related information is collected in the practice of the G.P., who plays a central role in the Dutch health care system.

## Conclusion

The population prevalence of treated thyroid disorders in Lifelines is 3.1%. In those not known to have a thyroid disorder, the prevalence of primary hypothyroidism is 0.7%, and of subclinical hypothyroidism is 9.4%. Furthermore, 0.7% has a suppressed TSH. Only 59.3% of those participants who use levothyroxine have a TSH level in the normal range. Reliable ascertainment of previous and new thyroid disorders is not possible based on the currently used questionnaires. The large group of individuals with subclinical hypothyroidism offers an excellent possibility to prospectively follow the natural course of this disorder. However, both structured questions as well as linking to G.P.'s and pharmacists' data are necessary to improve the completeness and reliability of data on thyroid disorders.

## Supporting information

**S1 Table. Baseline characteristics of the 152180 participants.**
(DOCX)

**S2 Table. Thyroid hormone levels measured at baseline in 39935 Lifelines participants.**
(DOCX)

**S3 Table. Distribution of TSH class per age group (in %) in NHANES participants not using thyroid hormone medication at baseline.**
(DOCX)

**S4 Table. Self-reported thyroid disorders or previous surgery using open questions at baseline.**
(DOCX)

**S5 Table. Self-reported thyroid disorders using open questions asking for other disorders during the follow-up questionnaires.**
(DOCX)

## Acknowledgments

The authors wish to acknowledge the services of the Lifelines Cohort Study, the contributing research centres delivering data to Lifelines, and all the study participants.

## Author Contributions

**Conceptualization:** Melanie M. van der Klauw, Bruce H. R. Wolffenbuttel.

**Formal analysis:** Hanneke J. C. M. Wouters, Bruce H. R. Wolffenbuttel.

**Investigation:** Anneke C. Muller Kobold.

**Resources:** Anneke C. Muller Kobold.

**Writing – original draft:** Hanneke J. C. M. Wouters, Bruce H. R. Wolffenbuttel.

**Writing – review & editing:** Hanneke J. C. M. Wouters, Sandra N. Slagter, Anneke C. Muller Kobold, Melanie M. van der Klauw, Bruce H. R. Wolffenbuttel.

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
