## [Decision Letter · Decision Letter 0]

27 Jul 2020

PONE-D-20-13620

Epidemiology of thyroid disorders in the Lifelines Cohort Study

PLOS ONE

Dear Dr. Wolffenbuttel, 

Thank you for submitting your manuscript to PLOS ONE. After careful consideration, we feel that it has merit but does not fully meet PLOS ONE’s publication criteria as it currently stands. Therefore, we invite you to submit a revised version of the manuscript that addresses the points raised during the review process.

We look forward to receiving your revised manuscript.

Kind regards,

Silvia Naitza

Academic Editor

PLOS ONE

Journal Requirements:

2. We note that Supporting Information file may include questionnaire items that may have been previously published in English. The reproduction of previously published work has implications for the copyright that may apply to these publications. We would be grateful if you could clarify whether you have obtained permission from the original copyright holder to republish these items under a CC BY license. If you have not obtained permission to publish these items please remove them from your manuscript. You may wish to replace the text you have removed with relevant question numbers/ brief descriptions of each item; please be sure to include any relevant references and in-text citations.

3.We note that you have indicated that data from this study are available upon request. PLOS only allows data to be available upon request if there are legal or ethical restrictions on sharing data publicly. For information on unacceptable data access restrictions, please see http://journals.plos.org/plosone/s/data-availability#loc-unacceptable-data-access-restrictions.

Additional Editor Comments (if provided):

Dear Dr. Wolffenbuttel,

we finally received the requested number of reviews for your manuscript from the two Referees, and we apologize for the long waiting. As you can see from the comments, we cannot publish your manuscript as it stands now, and we thus ask you to resubmit it after you have included the suggested changes, as well as a point-by-point response to their queries.

Best regards,

Silvia Naitza

Reviewers' comments:

Reviewer's Responses to Questions

**Comments to the Author**

1. Is the manuscript technically sound, and do the data support the conclusions?

Reviewer #1: Yes

Reviewer #2: Yes

2. Has the statistical analysis been performed appropriately and rigorously? 

Reviewer #1: Yes

Reviewer #2: I Don't Know

3. Have the authors made all data underlying the findings in their manuscript fully available?

Reviewer #1: Yes

Reviewer #2: Yes

4. Is the manuscript presented in an intelligible fashion and written in standard English?

Reviewer #1: Yes

Reviewer #2: Yes

5. Review Comments to the Author

Reviewer #1: The manuscript PONE-D-20-13620 is based on an analysis with the Lifelines Cohort Study from the North East region of the Netherlands. The geographical feature should be part of the title; example, "Epidemiology of thyroid disorders in the Lifelines Cohort Study (Netherland)"

Some comments should be included in the abstract (results section) on the results of the FT4 and FT3 levels. Since in the abstract (methods section) it is said “Data on baseline thyroid hormones measurements (TSH, FT4 and FT3) were available in a subset of 39935 participants”.

Figure 1 is not relevant for publication; your data is included several times in the text.

In figure 2, on the Y axis, it should say "Percentage using Levothyroxine" instead of "thyroid hormones", as its title describes.

Figure 3 does not include information on statistical analysis (model, correlation, p value ...), and a comment on this would be appropriate.

In Table 4, TSH units of measure are missing. In the last row the total number of participants with treatment is expressed and according to gender, it would be convenient to indicate “total”.

In table S2 the P-value should be indicated in a similar way to table S1.

The discussion should be structured, to improve its analysis, in the four objectives indicated in the abstract: a) prevalence of thyroid disorders, b) thyroid medication use, c) thyroid hormone levels, and d) validity of thyroid data obtained.

The discussion indicates “Our study has several strengths and weaknesses”. Part of these comments and others made in other sections of the manuscript should be included in a new section of the discussion on the limitations of the study.

References 28 and 38 of the manuscript must be completed with volume and / or pages of the publication; 30 (6): 821-828 and 9 (3): 792, respectively.

Reviewer #2: Comments to Author:

The study of Wouters et al. describes the epidemiology of thyroid disorders in the Lifelines Cohort Study. It aims at describing prevalence of thyroid disorders, thyroid medication use and thyroid hormone concentrations in the cohort. The manuscript is clear and concise, results are well described, also figures and tables.

I have few comments

- Line 179 : a reference should be added to ascertain the fact that the general Dutch population is iodine sufficient

- Line 179: anti-thyroid peroxidase Ab were not available: this sentence should be commented in method or discussion section. It is not a limitation since the cohort is large enough but precision should be noted.

- Line 189: unit is missing after fasting glucose level.

- Line 210: source of FT4 and FT3 reference interval should be added.

- Line 278: Participants using levothyroxine had higher FT4 and lower FT3. Time of blood sample is mentioned in method section, but it is not mentioned whether subjects took levothyroxine before or after blood sample. This result is not discussed in discussion and may be not necessary in the manuscript.

- Line 286: Figure 3 could be improved by mentioning on it the 98.4% of subjects with normal FT4 and TSH between 4 and 10mUI/L.

- Line 413: a study in France has been cited mentioning the need of age and sex specific reference values for TSH. Authors could have discussed this sentence in regard of their data: they mentioned that FT4 is normal in 98.4% of subjects with TSH between 4.0 and 10 mUI/L and that less than 60% of subject using levothyroxine have normal TSH. Also G.P. may have didifficulties to adjust levothyroxine medication.

6. PLOS authors have the option to publish the peer review history of their article (what does this mean?). If published, this will include your full peer review and any attached files.

Reviewer #1: No

Reviewer #2: No

---

## [Author Response · Author response to Decision Letter 0]

28 Aug 2020

Academic Editor:

We rechecked the style requirements and adapted the manuscript accordingly. 

2. We note that Supporting Information file may include questionnaire items that may have been previously published in English. The reproduction of previously published work has implications for the copyright that may apply to these publications. We would be grateful if you could clarify whether you have obtained permission from the original copyright holder to republish these items under a CC BY license. If you have not obtained permission to publish these items please remove them from your manuscript. You may wish to replace the text you have removed with relevant question numbers/ brief descriptions of each item; please be sure to include any relevant references and in-text citations.

In the Supporting Information 3 questionnaire items are described:

- HEALTH73: another disorder that you have not mentioned yet? (baseline)

- HEALTH74: previous thyroid surgery (baseline)

- HEALTH100: another disorder that you have not mentioned yet? (follow-up)

To the best of our knowledge, these questions were formulated by Lifelines and are not part of a previously published questionnaire. The general outline of the Lifelines dataset is publicly available on:

https://www.lifelines.nl/researcher/data-and-biobank/$6102/$6104

We have added this to the text.

The results presented in the manuscript are based on the baseline and follow-up measurement of the Lifelines Cohort dataset. Lifelines is the sole owner of any and all data and materials that has been collected from the participants. All data are stored and administered by Lifelines. The data are accessible to researchers through a virtual environment. Data is not publicly available due to restrictions imposed by the Lifelines Scientific Board and the Medical Ethical Committee of the University Medical Center Groningen related to protecting patient privacy and because the participants have not given permission for this. Lifelines is a facility which allows data access for reproducibility of the study results. Questions regarding the data availability can be addressed to the Research Office of Lifelines, data@research@lifelines.nl. We have added this information to the data availability statement.

We have removed this phrase as it is not a core part of our analyses.

Reviewer #1: 

The manuscript PONE-D-20-13620 is based on an analysis with the Lifelines Cohort Study from the North East region of the Netherlands. 

We cordially thank the reviewer for his / her comments.

The geographical feature should be part of the title; example, "Epidemiology of thyroid disorders in the Lifelines Cohort Study (Netherland)"

The authors would like to thank the reviewer for this comment. We agree that the geographical area should be part of the title of the current manuscript. We changed the title to ‘Epidemiology of thyroid disorders in the Lifelines Cohort Study (the Netherlands)’.

Some comments should be included in the abstract (results section) on the results of the FT4 and FT3 levels. Since in the abstract (methods section) it is said “Data on baseline thyroid hormones measurements (TSH, FT4 and FT3) were available in a subset of 39935 participants”.

We thank the reviewer for noting this. In the revised version we mention that over 98% of subjects with TSH between 4 and 10 mUI/L had normal FT4.

Figure 1 is not relevant for publication; your data is included several times in the text.

We have removed Figure 1. 

In figure 2, on the Y axis, it should say "Percentage using Levothyroxine" instead of "thyroid hormones", as its title describes.

We thank the reviewer for noting this. Accordingly, we corrected this in the revised version of the figure.

Figure 3 does not include information on statistical analysis (model, correlation, p value ...), and a comment on this would be appropriate.

We added information about the statistical analysis in the legend of the figure. 

In Table 4, TSH units of measure are missing. In the last row the total number of participants with treatment is expressed and according to gender, it would be convenient to indicate “total”.

We have corrected this. 

In table S2 the P-value should be indicated in a similar way to table S1.

We have corrected this.

The discussion should be structured, to improve its analysis, in the four objectives indicated in the abstract: a) prevalence of thyroid disorders, b) thyroid medication use, c) thyroid hormone levels, and d) validity of thyroid data obtained.

We thank the reviewer for this valuable comment. We now used the following headings to structure the discussion: 

A) prevalence of thyroid disorders based on use of thyroid medication

B) prevalence and incidence of thyroid disorders based on thyroid hormone levels

C) prevalence of abnormal TSH levels in individuals using levothyroxine

D) validity of the obtained thyroid data.

The discussion indicates “Our study has several strengths and weaknesses”. Part of these comments and others made in other sections of the manuscript should be included in a new section of the discussion on the limitations of the study.

We started a new section to describe the strengths and limitations of our study. In this section we referred to comments which were made previously in the text. 

References 28 and 38 of the manuscript must be completed with volume and / or pages of the publication; 30 (6): 821-828 and 9 (3): 792, respectively.

We have corrected this.

 

Reviewer #2:

The study of Wouters et al. describes the epidemiology of thyroid disorders in the Lifelines Cohort Study. It aims at describing prevalence of thyroid disorders, thyroid medication use and thyroid hormone concentrations in the cohort. The manuscript is clear and concise, results are well described, also figures and tables. I have few comments.

We cordially thank the reviewer for his / her comments.

- Line 179 : a reference should be added to ascertain the fact that the general Dutch population is iodine sufficient

We added the following reference: ‘Verkaik-Kloosterman J, Buurma-Rethans EJ, Dekkers AL, van Rossum CT. Decreased, but still sufficient, iodine intake of children and adults in the Netherlands. Br J Nutr. 2017;117(7):1020-1031.’

- Line 179: anti-thyroid peroxidase Ab were not available: this sentence should be commented in method or discussion section. It is not a limitation since the cohort is large enough but precision should be noted.

We have added a sentence on the usefulness of measuring anti-TPO antibodies in the discussion.

- Line 189: unit is missing after fasting glucose level.

We have corrected this.

- Line 210: source of FT4 and FT3 reference interval should be added.

We added that the references interval was based on local laboratory references values.

- Line 278: Participants using levothyroxine had higher FT4 and lower FT3. Time of blood sample is mentioned in method section, but it is not mentioned whether subjects took levothyroxine before or after blood sample. This result is not discussed in discussion and may be not necessary in the manuscript.

We agree with the reviewer that describing the difference in thyroid hormone levels between thyroid hormone users and non-users is not the goal of this study, but we consider this important information, also in the light of the last remark about difficulties GP’s may have to adjust levothyroxine medication. Also, the relevance of this has been recently illustrated by:

Gullo et al. Levothyroxine monotherapy cannot guarantee euthyroidism in all athyreotic patients. PLoS One 2011;6(8):e22552.

and

Flinterman et al. Impact of a forced dose-equivalent levothyroxine brand switch on plasma thyrotropin: a cohort study. Thyroid 2020;30(6):821-828. 

We have added this to the discussion.

- Line 286: Figure 3 could be improved by mentioning on it the 98.4% of subjects with normal FT4 and TSH between 4 and 10mUI/L.

We added in the legend of the figure that 98.4% of the individuals with a TSH between 4.01 and 10.0 mU/l had a normal FT4 (>11 pmol/l).

- Line 413: a study in France has been cited mentioning the need of age and sex specific reference values for TSH. Authors could have discussed this sentence in regard of their data: they mentioned that FT4 is normal in 98.4% of subjects with TSH between 4.0 and 10 mUI/L and that less than 60% of subject using levothyroxine have normal TSH. Also G.P. may have didifficulties to adjust levothyroxine medication.

We thank the reviewer for this valuable comment. The increasing upper limit of normal of TSH with aging, as presented in the study of Raverot et al., might partly explain the large number of individuals with a normal T4 in combination with a slightly elevated TSH as well as the large number of individuals treated with levothyroxine having an abnormal TSH. We extended the discussion section commenting on this.

---

## [Decision Letter · Decision Letter 1]

27 Oct 2020

PONE-D-20-13620R1

Epidemiology of thyroid disorders in the Lifelines Cohort Study (the Netherlands)

PLOS ONE

Dear Prof.  Wolffenbuttel,

Thank you for submitting your revised manuscript to PLOS ONE. After careful consideration, we feel that it has merit but does not fully meet PLOS ONE’s publication criteria as it currently stands. Therefore, we invite you to submit a revised version of the manuscript that addresses the points raised during the review process.

As you can see from the attached reviews, the referees are satisfied with your responses to their queries and have only indicated minor points to be amended. We are looking forward to receiving these final changes and proceed with the acceptance of your manuscript.

We look forward to receiving your revised manuscript.

Kind regards,

Silvia Naitza

Academic Editor

PLOS ONE

Additional Editor Comments (if provided):

Dear Prof.  Wolffenbuttel,

please find enclosed the reviews of the two referees on your revised manuscript PONE-D-20-13620. As you can see, they are satisfied with your responses to their queries and have only indicated minor points to be amended. We are looking forward to receiving these final changes and proceed with the acceptance of your manuscript.

Best regards,

Silvia Naitza

Reviewers' comments:

Reviewer's Responses to Questions

**Comments to the Author**

1. If the authors have adequately addressed your comments raised in a previous round of review and you feel that this manuscript is now acceptable for publication, you may indicate that here to bypass the “Comments to the Author” section, enter your conflict of interest statement in the “Confidential to Editor” section, and submit your "Accept" recommendation.

Reviewer #1: All comments have been addressed

Reviewer #2: All comments have been addressed

2. Is the manuscript technically sound, and do the data support the conclusions?

Reviewer #1: Yes

Reviewer #2: Yes

3. Has the statistical analysis been performed appropriately and rigorously? 

Reviewer #1: Yes

Reviewer #2: Yes

4. Have the authors made all data underlying the findings in their manuscript fully available?

Reviewer #1: Yes

Reviewer #2: Yes

5. Is the manuscript presented in an intelligible fashion and written in standard English?

Reviewer #1: Yes

Reviewer #2: Yes

6. Review Comments to the Author

Reviewer #1: Overall comments to the Author

Thank you for addressing my comments and for making edits to your paper.

However, in the manuscript of version R1, figures 2 and 3 modified according to the indications of the text are not included.

Reviewer #2: All comments have been adressed. The revised manuscript is clear.

The correct unit for TSH is mUI/L, this should be corrected.

7. PLOS authors have the option to publish the peer review history of their article (what does this mean?). If published, this will include your full peer review and any attached files.

Reviewer #1: No

Reviewer #2: No

---

## [Author Response · Author response to Decision Letter 1]

5 Nov 2020

Reviewer #1:

Thank you for addressing my comments and for making edits to your paper. However, in the manuscript of version R1, figures 2 and 3 modified according to the indications of the text are not included.

We thank the reviewer for his / her comment. In the submitted revised version of the manuscript we included 2 figures (‘Fig 1. Percentage of levothyroxine users per age group’ and ‘Fig 2. Relationship between TSH and FT4 in participants with subclinical or overt hypothyroidism not using levothyroxine’). As suggested we changed the Y axis label into ‘Percentage using Levothyroxine’ in Fig. 1 and added information about the statistical analysis in the legend of Fig 2. 

In the PDF proof, which we received during the submission process, the figures are on page 45 and 46. 

We hope that in this resubmission the figures are correctly shown and that we have informed you sufficiently. 

Reviewer #2:

All comments have been adressed. The revised manuscript is clear. The correct unit for TSH is mUI/L, this should be corrected.

We thank the reviewer for noting this. Accordingly, we corrected this in the revised version of the text and in Figure 2.

---

## [Editor Report · Decision Letter 2]

10 Nov 2020

Epidemiology of thyroid disorders in the Lifelines Cohort Study (the Netherlands)

PONE-D-20-13620R2

Dear Prof. Wolffenbuttel, 

We’re pleased to inform you that your manuscript has been judged scientifically suitable for publication and will be formally accepted for publication once it meets all outstanding technical requirements.

Kind regards,

Silvia Naitza

Academic Editor

PLOS ONE

Additional Editor Comments (optional):

Dear Prof. Wolffenbuttel,

I'm pleased to let you know that your revised manuscript PONE-D-20-13620R2 as it stands now has addressed all the reviewers'requests and is therefore suitable for publication in Plos One.

Best regards,

Silvia Naitza
---

## [Editor Report · Acceptance letter]

16 Nov 2020

PONE-D-20-13620R2 

Epidemiology of thyroid disorders in the Lifelines Cohort Study (the Netherlands) 

Dear Dr. Wolffenbuttel:

I'm pleased to inform you that your manuscript has been deemed suitable for publication in PLOS ONE. Congratulations! Your manuscript is now with our production department. 

Kind regards, 

on behalf of

Dr. Silvia Naitza 

Academic Editor

PLOS ONE